

# Taxonomic description and phylogenetic placement of two new species of *Spalangiopelta* (Hymenoptera: Pteromalidae: Ceinae) from Eocene Baltic amber

Marina Moser[1,2], Roger A. Burks[3], Jonah M. Ulmer[1,2], John M. Heraty[3], Thomas van de Kamp[4,5] and Lars Krogmann[1,2]

[1] Department of Entomology, State Museum of Natural History Stuttgart, Stuttgart, Germany
[2] Institute of Biology, Systematic Entomology (190n), University of Hohenheim, Stuttgart, Germany
[3] Department of Entomology, University of California Riverside, Riverside, CA, USA
[4] Institute for Photon Science and Synchrotron Radiation (IPS), Karlsruhe Institute of Technology (KIT), Eggenstein-Leopoldshafen, Germany
[5] Laboratory for Applications of Synchrotron Radiation, Karlsruhe Institute of Technology (KIT), Karlsruhe, Germany

## ABSTRACT

*Spalangiopelta* is a small genus of chalcid wasps that has received little attention despite the widespread distribution of its extant species. The fossil record of the genus is restricted to a single species from Miocene Dominican amber. We describe two new fossil species, *Spalangiopelta darlingi* sp. n. and *Spalangiopelta semialba* sp. n. from Baltic amber. The species can be placed within the extant genus *Spalangiopelta* based on the distinctly raised hind margin of the mesopleuron. 3D models reconstructed from µCT data were utilized to assist in the descriptions. Furthermore, we provide a key for the females of all currently known *Spalangiopelta* species. The phylogenetic placement of the fossils within the genus is analyzed using parsimony analysis based on morphological characters. Phylogenetic and functional relevance of two wing characters, admarginal setae and the hyaline break, are discussed. The newly described Baltic amber fossils significantly extend the minimum age of *Spalangiopelta* to the Upper Eocene.

## INTRODUCTION

Ceinae is a small subfamily of Pteromalidae within the megadiverse Chalcidoidea. The subfamily contains three genera, two of which are monotypic (*Mitroiu, 2016*): *Bohpa* Darling, 1991, which is known only from South Africa, and *Cea* Walker 1837, which is found throughout the Western Palearctic region. The genus *Spalangiopelta* Masi 1922 comprises 14 extant species, eight of which are known only from the Palearctic region, whereas the other six species are Nearctic or Neotropic (*Darling, 1991*; *Mitroiu, 2016*). In addition, *Darling (1996)* described *Spalangiopelta georgei*, a fossil from Dominican amber.

Corresponding authors
Marina Moser,
marina.moser@smns-bw.de
Lars Krogmann,
lars.krogmann@smns-bw.de

Specimens of Ceinae are rarely represented in collections (*Darling, 1991*). To date, the biology and life history strategies of members of the subfamily are virtually unknown with the exception of two host records: *Spalangiopelta alata* Bouček 1953 was observed emerging from a leaf mine of the herbivorous drosophilid *Scaptomyza flava* (Fallen, 1823) (Diptera: Drosophilidae, originally cited as *Scaptomyza flaveola* Meigen 1830) on *Cakile maritima* Scopoli (Brassicaceae) (*Bouček, 1961*). In addition, *Cea pulicaris* Walker 1837 was reared from *Phytomyza pauliloewii* Hendel 1920 (Diptera: Agromyzidae) on *Peucedanum oreoselinum* (L.) Moench (Apiaceae) (*Bouček, 1961*). The morphology of the mesosoma, which displays either a distinctly arched or a flattened configuration led to the assumption that *Spalangiopelta* females are associated with leaf litter and duff habitats (*Darling, 1991*). From these findings it would seem that leaf-mining Diptera living in confined habitats are the hosts of *Spalangiopelta*.

At the time of description, the Dominican amber fossil *S. georgei* was dated to the Oligocene based on an age estimate of 23 and 30 million years (*Darling, 1996*; *Grimaldi, 1995*). However, more recent estimates on the age of Dominican amber reach from 15 to 20 million years with the highest density of resin-producing trees in the Miocene 16 million years ago (*Iturralde-Vinent & MacPhee, 1996*; *Iturralde-Vinent, 2001*). This reduces the previously assumed minimum age of *Spalangiopelta* considerably.

Baltic amber was formed during the Eocene but there is still considerable debate concerning its exact age and origin. Equally, the botanical origin of Baltic amber has remained the subject of scientific debate for decades and was portentously dubbed "The Tertiary Baltic Amber Mystery" (*Langenheim, 2003*). Conservative studies build upon an allochthonous redeposition of Baltic amber into layers of Blaue Erde (Blue Earth) and therefore deduce an age at least matching that of the surrounding layers. Following this assumption, estimates of the age of Baltic amber range from 35 to 55 million years (*Poinar, 1992* and references therein) with several authors giving a narrower range of 40 to 47 million years (*Burleigh & Whalley, 1983*; *Grimaldi, 1995*; *Grimaldi, 1996*; *Ritzkowski, 1997*). In contrast, another theory has emerged in recent years: based on the hypothesis that locality and time of the formation of Baltic amber do not differ significantly from its deposition in marine sediments, an age for Baltic amber of 35 to 43 million years was inferred (*Sadowski et al., 2017*; *Standke, 2008*).

The current study addresses two fossil specimens of *Spalangiopelta* (Chalcidoidea: Ceinae) from Eocene Baltic amber, which fall within the known geographic range of extant Palearctic *Spalangiopelta* species. The study has the following objectives: (1) Taxonomic description of two fossil wasp specimens from Baltic amber, complemented by digital images, scientific illustrations and 3D models reconstructed from CT data. (2) Development of a morphological key to enable the identification of extant and fossil *Spalangiopelta* species. (3) Phylogenetic placement of the two fossil specimens based on the cladistic analysis of morphological characters. As the oldest fossils of *Spalangiopelta* described so far, the two fossils extend the minimum age of *Spalangiopelta* to the upper Eocene and thus more than double the minimum age of the genus.

**Table 1** Data matrix for *Spalangiopelta* females and the outgroup *Cea pulicaris*.

| Character | 1 | 2 | 3 | 4 | 5 | 6 | 7 | 8 | 9 | 10 | 11 | 12 | 13 | 14 | 15 | 16 | 17 | 18 | 19 | 20 | 21 | 22 |
|---|---|---|---|---|---|---|---|---|---|---|---|---|---|---|---|---|---|---|---|---|---|---|
| Character number in *Darling (1996)* | 1 | 2 | 3 | 4 | 7 | 8 | 9 | 10 | 11 | 12 | 13 | 14 | 15 | 16 | 17 | 18 | 19 | 20 | 21 | 23 | 24 | – |
| *Cea pulicaris* | 0 | 0 | 0 | 0 | 0 | 0 | 0 | 0 | 0 | 0 | 0 | 0 | 0 | 0 | 0 | 0 | 0 | 0 | 0 | 0 | 0 | 0 |
| *S. alata* | 1 | 1 | 0 | 1 | 1 | 1 | 1 | 3 | 0 | 1 | 1 | 0 | 0 | 2 | 1 | 1 | 0 | 1 | 0 | 3 | 1 | 1 |
| *S. albigena* | 1 | 0 | 2 | 4 | 1 | 0 | 1 | 1 | 1 | 2 | 1 | 1 | 1 | 3 | 0 | 1 | 0 | 1 | 1 | 2 | 0 | 1 |
| *S. alboaculeata* | 0 | 1 | 0 | 3 | ? | 0 | 0 | 1 | 0 | 0 | 0 | 0 | 0 | 2 | 0 | 1 | 0 | 1 | 0 | 1 | 0 | 1 |
| *S. apotherisma* | 0 | 0 | 0 | 0 | 1 | 1 | 0 | 1 | 0 | 0 | 0 | 0 | 0 | 2 | 1 | 1 | 1 | 1 | 0 | 1 | 0 | 1 |
| *S. brachyptera* | 0 | 1 | 0 | 0 | 0 | – | 0 | 1 | 0 | 0 | 0 | 0 | 0 | 2 | 0 | – | – | 1 | – | 1 | 0 | 0 |
| *S. canadensis* | 1 | 1 | 0 | 1 | 1 | 0 | 1 | 3 | 0 | 1 | 1 | 0 | 0 | 2 | 1 | 1 | 0 | 1 | 0 | 3 | 1 | 1 |
| *S. ciliata* | 0 | 0 | 1 | 2 | 1 | 2 | 1 | 1 | 0 | 0 | 0 | 0 | 1 | 1 | 0 | 1 | 0 | 1 | ? | 1 | 0 | 0 |
| *S. dudichi* | 0 | 0 | 0 | 0 | 1 | 1 | 0 | 1 | 0 | 0 | 0 | 0 | 0 | 2 | 1 | 1 | 1 | 1 | 0 | 1 | 0 | 1 |
| *S. felonia* | 0 | 1 | 0 | 0 | 0 | 1 | 0 | 1 | 0 | 0 | 0 | 0 | 0 | 2 | 0 | 1 | 0 | 1 | 0 | 1 | 0 | 1 |
| *S. georgei* | 1 | 0 | 0 | 4 | ? | 1 | 0 | 2 | 0 | 2 | 1 | 0 | 0 | ? | ? | 1 | 0 | 2 | 2 | 3 | 0 | 1 |
| *S. hiko* | 0 | 0 | 0 | 1 | 1 | 1 | 0 | 1 | 0 | 1 | 1 | 0 | 0 | 2 | 1 | 1 | 1 | 1 | 0 | 1 | 0 | 1 |
| *S. laevis* | 1 | 0 | 0 | 4 | ? | 0 | 1 | 1 | 1 | 2 | 1 | 1 | 0 | 3 | 1 | 1 | 0 | 2 | 1 | 2 | 0 | 1 |
| *S. procera* | 0 | 0 | 0 | 1 | ? | 1 | 0 | 2 | 0 | 2 | 0 | 0 | 0 | 2 | 0 | 1 | 1 | 1 | 0 | 1 | 0 | 1 |
| *S. rameli* | 0 | 1 | 0 | 0 | 1 | 1 | 0 | 2 | 0 | 0 | 0 | 0 | 0 | 4 | 0 | 1 | 1 | 1 | 0 | 1 | 0 | 1 |
| *S. viridis* | 0 | 1 | 0 | 0 | 1 | 1 | 0 | 1 | 0 | 1 | 0 | 0 | 0 | 4 | 0 | 1 | 0 | 1 | 0 | ? | 0 | 1 |
| *S. darlingi* sp. n. | 0 | 0 | 3 | 0 | ? | 1 | 0 | 1 | 0 | 2 | 1 | 0 | 0 | 1 | 0 | 1 | 1 | 1 | 0 | 1 | 0 | 1 |
| *S. semialba* sp. n. | ? | 0 | 0 | 4 | ? | 1 | ? | 1 | 0 | ? | 1 | 1 | 0 | ? | ? | 1 | 1 | 1 | 0 | 1 | 0 | 1 |

**Notes.**

0, ancestral state; 1, 2, 3, 4, derived states; ?, missing data; - =, gaps.

See text and *Darling (1996)* for description of characters and states.

## MATERIALS & METHODS

### Specimen description and terminology

Both holotypes are deposited at the State Museum of Natural History Stuttgart (SMNS). The morphological terminology in this study follows that of the Hymenoptera Anatomy Ontology (*Yoder et al., 2010*) with some additional terms from *Darling (1991)*. Terms relating to surface sculpturing are based on *Harris (1979)*. The description format largely follows *Darling (1996)*, allowing comparison between the amber fossils of *Spalangiopelta*.

### Morphological description and imaging

Observations, descriptions and scoring of morphological characters (Table 1) were compiled using a Leica M205C stereo microscope with a 7.8 to 160× magnification. The illustration of *Spalangiopelta darlingi* sp. n. was made with the same equipment with a camera lucida attached. A MZ 16 APO Leica R microscope with an attached DXM 1200 Leica R camera was used for habitus imaging with subsequent stacking of images in Helicon focus version 7.6.1 (Helicon Soft Ltd, Kharkov, Ukraine). Stacking followed the pyramid approach (method C) with the smoothing parameter setting of 4 to reduce image artefacts. Detail images were taken with a Keyence VHX 5000 digital microscope. Exact measurements given in Table 2 were taken in Amira based on the 3D models. All images

**Table 2 Amira measurements of structures of *Spalangiopelta darlingi* sp. n. and *Spalangiopelta semialba* sp. n., holotypes.** All measurements are in μm.

| Measurement | *Spalangiopelta darlingi* sp. n. | *Spalangiopelta semialba* sp. n. |
|---|---|---|
| Total body length | 742 | 550 |
| Length of head | 116 | 103 |
| Length of mesosoma | 253 | 231 |
| Length of metasoma | 342 | 220 |
| T2 | 77 | 26 |
| T3 | 21 | 28 |
| T4 | 53 | 26 |
| T5 | 64 | 30 |
| T6 | 71 | 42 |
| T7 | 55 | 48 |
| T8 | 35 | 21 |
| Ovipositor sheath length | 134 | 154 |
| Ovipositor length | 111 | 90 |
| HH | 198 | 163 |
| HW | 195 | 86 |
| Eye height | 124 | 72 |
| Eye breadth | 117 | 69 |
| POL | 63 | 50 |
| OOL | 29 | 26 |
| LOL | 30 | — |
| Malar sulcus | 43 | 40 |
| Distance between clypeus and radicle | 32 | 30 |
| Mandible length | 49 | – |
| Mandible height | 19 | – |
| Radicle | 15 | 9 |
| Scape | 104 | 62 |
| Pedicellus | 39 | 42 |
| Total length of anelli | 12 | 10 |
| F1 | 14 | 8 |
| F2 | 21 | 10 |
| F3 | 24 | 18 |
| F4 | 25 | 24 |
| F5 | 28 | 17 |
| Clava (C1-C3) | 105 | 79 |
| Forewing length | 550 | 449 |
| Forewing breadth | 215 | 156 |
| Hindwing length | 464 | 368 |
| Hindwing breadth | 72 | 41 |
| Fore coxa length | 94 | 50 |
| Fore trochanter length | 17 | 24 |

**Table 2** (*continued*)

| Measurement | *Spalangiopelta darlingi* sp. n. | *Spalangiopelta semialba* sp. n. |
| --- | --- | --- |
| Fore femur length | 135 | 103 |
| Fore tibia length | 110 | 92 |
| Fore tibial spur length | 18 | – |
| Fore tarsus length | 105 | 107 |
| Mid coxa length | 60 | 35 |
| Mid trochanter length | 22 | 26 |
| Mid femur length | 134 | 100 |
| Mid tibia length | 157 | 125 |
| Mid tibial spur length | 15 | 15 |
| Mid tarsus length | 102 | 123 |
| Hind coxa length | 116 | – |
| Hind trochanter length | 35 | 25 |
| Hind femur length | 139 | 106 |
| Hind tibia length | 176 | 147 |
| Hind tibial spur length | 13 | – |
| Hind tarsus length | 118 | 129 |
| Prepectus height | 77 | 36 |
| Prepectus breadth | 49 | 27 |

were processed and figures were assembled in Adobe Photoshop version CS5.1 (Adobe Systems Software Ireland Ltd, Dublin, Ireland).

## μCT reconstruction

Synchrotron-based X-ray microtomography was performed at the imaging cluster of the KIT light source at Karlsruhe Institute of Technology, Germany. We used a parallel polychromatic X-ray beam produced by a 1.5 T bending magnet that was spectrally filtered by 0.5 mm aluminium. A fast indirect detector system was employed, consisting of a 12 m LSO:Tb scintillator (*Cecilia et al., 2011*), diffraction-limited optical microscope (Optique Peter) coupled with a 12 bit pco.dimax high speed camera with 2016 × 2016 pixels. Scans were done by taking 3,000 projections at 70 fps and an optical magnification of 10X, resulting in an effective pixel size of 1.22 μm. Tomographic reconstruction was performed by the UFO framework (*Vogelgesang et al., 2012*). Reconstruction of the 3D surface model followed the methodology described by *Ruthensteiner & Hess (2008)* and *van de Kamp et al. (2018)*. The tomogram images were imported into Amira 6.5.0 (Thermo Fisher Scientific, Waltham, MA, USA) and preliminary volume renderings were created, giving an overview of the fossils. The models were segmented with the segmentation editor in Amira by labelling every tenth slice. The labels served as input for subsequent semi-automatic segmentation using the software "Biomedisa" (*Lösel et al., 2020*). For the final models, Biomedisa's smoothed and filled segmentation results were chosen. The images of the 3D models presented in this study are screenshots of the surface models that were constructed as outlined above.

## Phylogenetic analysis and taxon sampling

Phylogenetic analyses were carried out based on the character matrix of *Darling (1996)*, which comprises 24 morphological characters. The characters for the new fossils were scored and added. Character states of *Spalangiopelta rameli* Mitroiu, 2016 and *Spalangiopelta viridis* Mitroiu, 2016 were retrieved from literature and added as the respective species description allowed (Table 1). Altogether, the phylogenetic analysis in the present study includes 17 ingroup species (14 extant species and three fossil species), representing all known species of *Spalangiopelta* (Table 1). *C. pulicaris* was used as outgroup because of the morphological similarity and numerous shared characters with its sister genus *Spalangiopelta*. Character states were assigned to the respective *Spalangiopelta* species based on (re-)descriptions in the revision by *Darling (1991)* or the original descriptions (*Darling, 1995*; *Mitroiu, 2016*; *Walker, 1851*).

*Character 1*: Hyaline break in parastigma. 0: absent, 1: present. The third character state ("partial") in *Darling (1996)* was removed, returning the character to a binary state. The partial hyaline break in *S. georgei* was assigned to state 1 because it is present regardless of its incompleteness.

*Character 2*: Length of ovipositor sheaths. 0: long, sheaths extended beyond apex of metasoma at least one-half length of hind tibia. 1: short, sheaths only slightly protruding, at most one-third length of hind tibia.

*Character 3*: Sculpture of midlobe of mesoscutum. 0: imbricate, 1: longitudinally striate, 2: transversely costulate, 3: alveolate. The third character state ("alveolate") was added to classify *Spalangiopelta darlingi* sp. n. adequately.

*Character 4*: Length of petiole. 0: transverse, length about one-half maximum width, 1: campanulate, slightly longer than wide, 2: elongate, length about twice maximum width, 3: subquadrate, 4: inconspicuous.

*Character 5*: Presence of short-winged females. 0: present, 1: absent. This character is classified as "present" in three species, *Spalangiopelta felonia* and *Spalangiopelta brachyptera*, as well as the outgroup *C. pulicaris*. In these species, some female specimens have wings so short that they do not reach the middle of the propodeum. It should be pointed out that both macropterous and brachypterous females are known in *S. felonia* and *C. pulicaris*, whereas in *S. brachyptera* all females have extremely reduced fore wings. Data were coded as "missing" for six species (*Spalangiopelta alboaculeata*, *Spalangiopelta hiko*, *Spalangiopelta laevis* and the fossil species *Spalangiopelta georgei*, *Spalangiopelta darlingi* sp. n. and *Spalangiopelta semialba* sp. n.) because as yet the number of specimens available is too low to assess with certainty if short-winged females are present in the six species in question.

*Character 6*: Shape of hind margin of fore wing. 0: Fore and hind margins of fore wing parallel, 1: Fore wing margins not parallel, hind margin rounded, 2: Hind margin narrowed toward apex.

*Character 7*: Admarginal setae along the marginal vein. 0: present, 1: absent. The number of admarginal setae varies within individuals of the same species, therefore this character only codes for presence or absence of admarginal setae but not for the exact number of setae (*Darling, 1991*).

*Character 8*: Shape of funicular segments of antenna. 0: funicular segments very long, length at least three times width, 1: quadrate, length subequal to width, 2: elongate, length about twice width, antenna overall slender with funicular segments cylindrical, 3: elongate, length about twice width, antenna overall robust with funicular segments widest in the middle, slightly barrel-shaped.

*Character 9*: Scape shape. 0: linear, cylindrical, 1: expanded, appears more robust. In the newly described fossil species, this character is reconstructible only through CT scans as the scape is hardly visible under the microscope.

*Character 10*: Sculpture of petiole. 0: with longitudinal costae, 1: alveolate without longitudinal costae, 2: smooth, without longitudinal costae.

*Character 11*: Size of metascutellum. 0: short, length much less than one-half length of mesoscutellum along midline. 1: long, about one-half length of mesoscutellum along midline, longer than frenum.

*Character 12*: Colour pattern of head. 0: concolourous with mesopleuron, 1: bicoloured, dark with white colouration on gena. *Spalangiopelta semialba* sp.n. is best described by character state 1 ("bicoloured") based on the bright colouration of the genal space. This bicoloured pattern does not seem to be an artefact of fossilisation due to the consistent presence on both sides of the head.

*Character 13*: Colour of fore coxa. 0: concolourous with mesopleuron, 1: white, strongly contrasted with mesopleuron.

*Character 14*: Sculpture of sub-median areas of propodeum. 0: imbricate, sculpture partly overlapping, appearing scaled, 1: finely areolate, divided into several small, irregular spaces, 2: alveolate, 3: glabrous, smooth, without any sculpture.

*Character 15*: Median carina on propodeum. 0: absent, 1: present.

*Character 16*: Angle of marginal and submarginal vein. 0: parallel, 1: angled. The outgroup *C. pulicaris* is the only species with character state 0, therefore this character does not provide information on the relationships within the genus *Spalangiopelta*.

*Character 17*: Shape of fore wing. 0: narrow, length greater than 3.5 times width, 1: wide, length less than 3.5 times width.

*Character 18*: Length of stigmal vein. 0: very short, almost sessile, 1: longer, about one-half length of postmarginal vein, PM/SV = 1.8–3.5, 2: very long, almost as long as postmarginal vein, PM/SV = 1.3–1.5.

*Character 19*: Angle of stigmal vein. 0: 45 degree angle with marginal and postmarginal veins, 1: 35 degree angle with marginal and postmarginal veins, 2: 25 degree angle with marginal and postmarginal veins. The angle of the stigmal vein with the marginal and postmarginal veins appeared slightly wider than 45 degrees in *S. semialba* sp.n., however due to the rather poor preservation of the wing venation and the resulting difficulty in measuring the angle accurately, it is classified as character state 0 ("45 degree").

*Character 20*: Size of radicle. 0: very long, about one-third scape length. 1: short, only about one-tenth scape length, 2: robust, about one-fifth scape length, 3: long, about one-fourth scape length. The robust appearance of the radicle in *S. albigena* and *S. laevis* is likely due to their expanded, unusually robust scape (see Character 9).

*Character 21*: Configuration of notauli. 0: distinct and linear, 1: shallow, broadly concave at scutellum, 2: shallow and indicated as broad pit-shaped depressions at mesoscutellum.

*Character 22*: Malar sulcus. 0: absent. 1: present Note that the malar sulcus is present in 15 species of *Spalangiopelta*, yet in the key to the subfamilies of Pteromalidae *Graham (1969)* lists "Malar sulcus absent" as one of the defining characters of Ceinae.

Mesquite version 3.61 (*Maddison & Maddison, 2019*) was used to assemble and modify the character matrix and for subsequent tree view and character tracing. Parsimony analyses were carried out in TNT version 1.5 (*Goloboff, Farris & Nixon, 2008*; *Goloboff & Catalano, 2016*) with RAM usage set to 500 Mbytes and maximum space for 10,000 trees in memory. The small size of the data set allowed for the use of traditional search and implicit enumeration in TNT with *C. pulicaris* designated as outgroup. Traditional search was conducted with 100,000 replications and tree bisection reconnection (TBR) with 1,000 trees saved per replication. Initially, all characters were unweighted. A strict consensus tree was calculated of the trees retained from traditional search and implicit enumeration. For implied weighting, a succession of concavity functions ranging from $k = 1$ to $k = 12$ were tested. Consistency indices and retention indices were calculated with the TNT script "STATS". Bremer support was calculated using PAUP version 4.0a166 (*Swofford, 2003*).

## LSID registration

The electronic version of this article in Portable Document Format (PDF) will represent a published work according to the International Commission on Zoological Nomenclature (ICZN), and hence the new names contained in the electronic version are effectively published under that Code from the electronic edition alone. This published work and the nomenclatural acts it contains have been registered in ZooBank, the online registration system for the ICZN. The ZooBank LSIDs (Life Science Identifiers) can be resolved and the associated information viewed through any standard web browser by appending the LSID to the prefix http://zoobank.org/. The LSID for this publication is: urn:lsid:zoobank.org:pub:F3A4D890-3480-41D2-B5A4-7C18E047C920. The online version of this work is archived and available from the following digital repositories: PeerJ, PubMed Central and CLOCKSS.

## RESULTS

### Key to females of extant and fossil *Spalangiopelta* species

1. Ovipositor sheaths only slightly extended beyond apex of metasoma, protruded distance less than 0.5 length of hind tibia.............................................................**2**

—Ovipositor sheaths distinctly extended beyond apex of metasoma, protruded distance more than 0.5 length of hind tibia ............................................................**8**

2. Fore wing with transparent region ("hyaline break") in parastigma. Admarginal setae below marginal vein absent.......................................................**3**

—Fore wing without transparent region ("hyaline break") in parastigma. Admarginal setae below marginal vein present and large .................................................**4**

3. Notauli shallow and linear, broadly concave at mesoscutellum. Hind margin of fore wing parallel to fore margin. Length of metascutellum much less than half of mesoscutellum along midline ................................................*S. canadensis* Darling, 1991

—Notauli as broad, pit-like depressions. Fore wing rounded with hind margin not parallel to fore margin. Metascutellum longer than frenum, about half length of mesoscutellum along midline ................................................*S. alata* Bouček, 1953

4. Funicular segments of antenna robustly elongate, length about 2 times width..........**5**

—Funicular segments of antenna subquadrate ................................................**6**

5. Fore wing narrow, length greater than 3.4 times width. Sculpture of petiole alveolate................................................*S. viridis* Mitroiu, 2016

—Fore wing wide, length 2.8 times maximum width. Petiole with longitudinal costae................................................*S. rameli* Mitroiu, 2016

6. Malar sulcus distinct ................................*S. felonia* Darling & Hanson, 1986

—Malar sulcus absent................................................**7**

7. First funicular segment quadrate, length subequal to width. Petiole transverse. Ovipositor dark, not distinctly white................................................*S. brachyptera* Masi, 1922

—First funicular segment elongate, length about 1.5 width. Petiole subquadrate. Ovipositor distinctly white ................................................*S. alboaculeata* Darling, 1995

8. Wings completely hyaline, without any maculation. Known only from Baltic amber................................................**9**

—Wings with maculation, at least faintly infuscate marks................................**10**

9. Midlobe of mesoscutum coarsely alveolate. Head concolourous, genae not strongly contrasted ................................................*Spalangiopelta darlingi* Moser sp. n. †

—Midlobe of mesoscutum superficially imbricate. Colour pattern of head bicoloured, brown with white colouration on genae ............*Spalangiopelta semialba* Moser sp. n. †

10. Fore wing narrow, length greater than 3.5 width ................................................**11**

—Fore wing wide, length about 3 times maximum width................................**14**

11. Fore coxa white, strongly contrasted with mesopleuron................................**12**

—Fore coxa brown to black, concolourous with mesopleuron................................**13**

12. Petiole elongate, length about twice maximum width. Head concolourous. Hind margin of fore wing narrowed toward apex. Scape linear. Submedian areas of propodeum finely areolate. Malar sulcus absent................................................*S. ciliata* Yoshimoto, 1997

—Petiole inconspicuous. White colouration posterior of genae. Hind and fore margin of fore wing parallel, with asetose region along hind margin of fore wing. Scape expanded. Submedian areas of propodeum glabrous. Malar sulcus present and located posteriorly................................................*S. albigena* Darling, 1991

13. Colour pattern of head bicoloured, brown with white colouration on genae. Fore wing with hyaline break in parastigma. Scape expanded. Funicular segments of antenna subquadrate. Stigmal vein angle 35° ................................................*S. laevis* Darling, 1991

—Head concolourous, genae not strongly contrasted. Fore wing with admarginal setae. Scape linear. Stigmal vein angle 25°. Fossil known from Dominican amber ................................................*S. georgei* † Darling, 1997

14. Median carina on propodeum distinct................................................**15**

—Median carina on propodeum absent.…………………………………………………**16**
15. Petiole transverse, with longitudinal costae. Body colour dark blue-violet with iridescent reflections ……………………………………………………***S. dudichi*** Erdös, 1955
—Petiole campanulate, slightly longer than wide, alveolate. Body colour iridescent blue–green ……………………………………………………***S. hiko*** Darling, 1995
16. Funicular segments subquadrate. Petiole transverse with longitudinal costae. Body colour blue–green ……………………………***S. apotherisma*** Darling & Hanson, 1986
—Funicular segments slenderly elongate, length about twice width. Petiole campanulate, slightly longer than wide, smooth. Body colour dark blue-violet with iridescent reflections.…………………………………………………***S. procera*** Graham, 1966

## Species description of *Spalangiopelta darlingi* Moser, sp. n.

(Figs. 1–3)
LSID: urn:lsid:zoobank.org:act:91C3491D-167E-438E-A336-36534722525D

Holotype: ♀, Eocene Baltic amber, collection number "SMNS BB-2851".
Type repository: SMNS.
    The specimen is located close to the edge of the amber piece. The left side of the specimen is well-preserved. On the right side, the metasoma is dented dorsally and therefore slightly deformed.
    Exact provenance unknown. Specimen is from the Krylov collection, purchased by the SMNS in 2008.
Distribution: Known only from the holotype.
Etymology: The specific name is a patronym for D. Christopher Darling to honour his work on parasitoid Hymenoptera and without whose contributions the genus *Spalangiopelta* would lack half of its described species.
Diagnosis: *Spalangiopelta darlingi* is characterized by the distinct alveolate sculpture on the mesoscutum. It is the only species with the following combination of characters: long ovipositor sheaths (protruded beyond apex of metasoma more than 0.5 length of hind tibia), fore wing with six admarginal setae and no hyaline break, petiole transverse and smooth (without longitudinal costae). *Spalangiopelta darlingi* is one of two known Eocene *Spalangiopelta* amber fossils. It can readily be distinguished from the other Baltic amber fossil *S. semialba* by the concolourous head and from the Dominican amber fossil *S. georgei* by the absence of a hyaline break in the anterior part of the marginal vein, the greater angle of the stigma vein with the anterior margin of the fore wing (45° rather than 25°) and by the completely hyaline wings.
Description: Female. Length, about 0.75 mm. Head concolourous dark brown with blue–green iridescent reflections, antennae dark brown. Mesosoma, coxae and femora

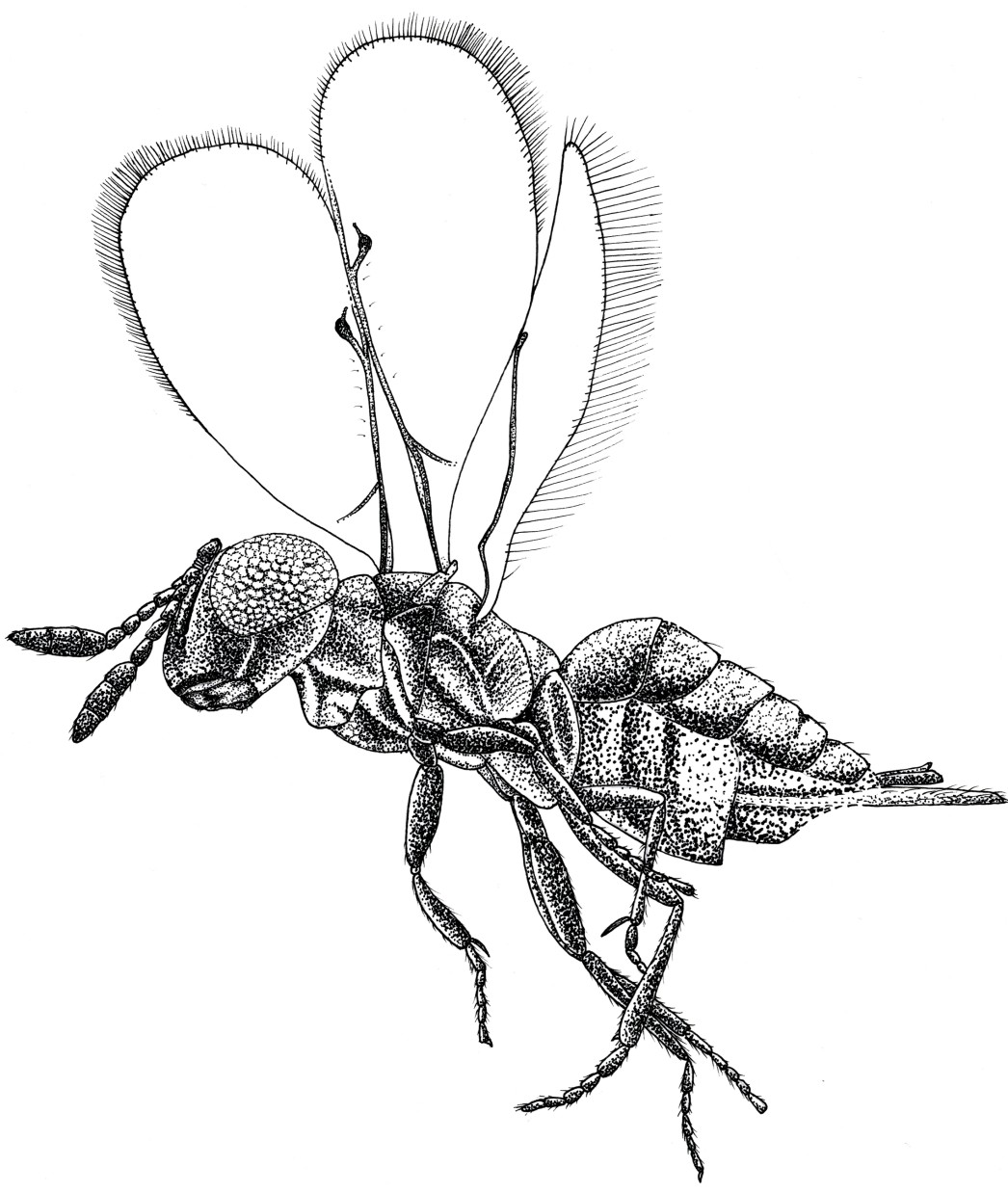

**Figure 1** **Habitus drawing of *Spalangiopelta darlingi* sp. n., holotype, female.** Lateral left side.

concolourous with head, hind tibia and tarsus slightly lighter. Fore wing long, extended beyond apex of metasoma 0.7 times body length, hyaline. Metasoma reddish-brown.

*Head*: In frontal view quadrate, HW/HH: 1.0. Sculpture of head weakly imbricate, except vertex finely reticulate. Black setae present along the parascrobal area and on the interorbital plane, vertex and ocular-ocellar areas devoid of pubescence. Malar sulcus distinct and

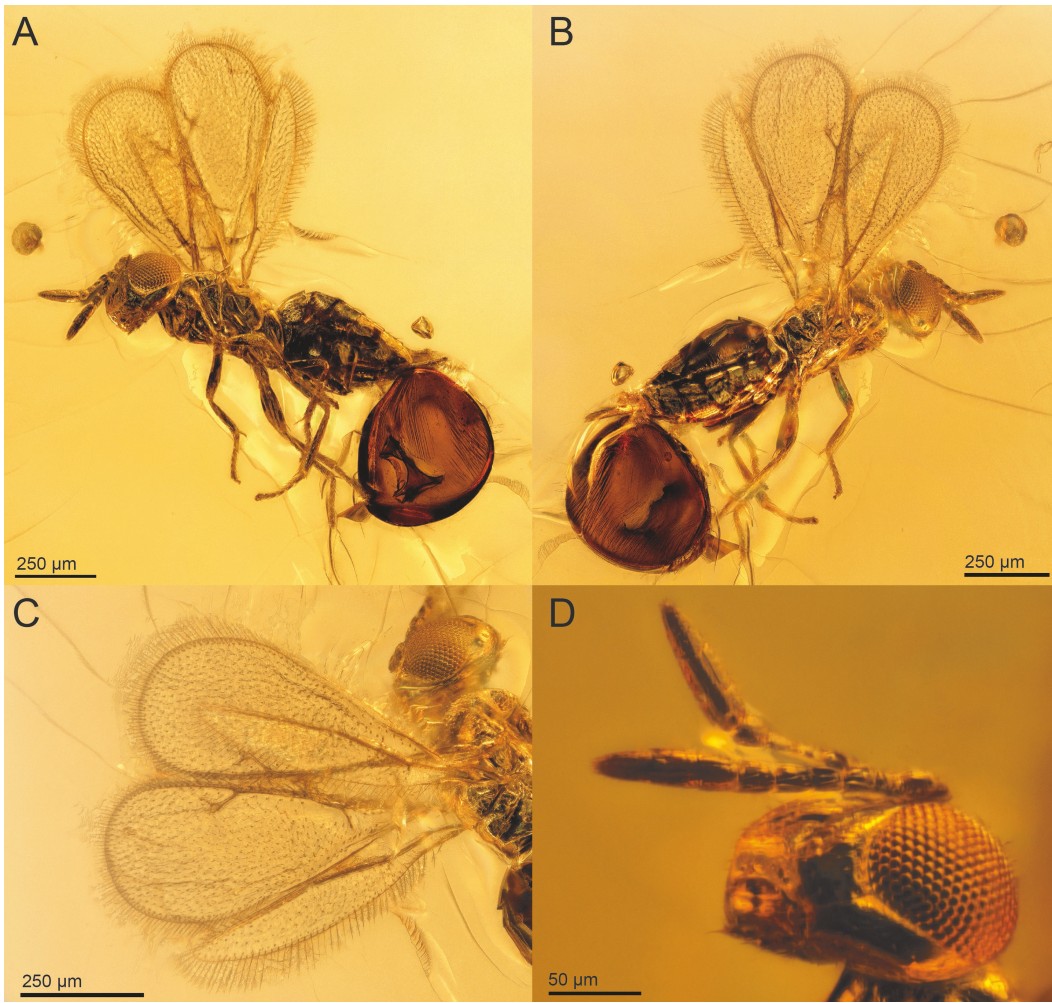

**Figure 2** *Spalangiopelta darlingi* **sp. n., holotype, female.** (A) Lateral habitus of the left side. (B) Lateral habitus of the right side. (C) Detail image of wings. (D) Detail image of head with antennae.

located more posterior than in other species of *Spalangiopelta*, one-third eye height. OOL about 0.5 POL.

Antenna: radicle very short, 0.14 scape length. Scape slightly arched, linear. A3 elongate, slightly wider than long, longer than A1 + A2. Length of F1 shorter than width; F2-F5 subequal in length, slightly elongate, length 1.2 times width, segments conical. Clava elongate, subequal in length to F1-F5 combined, equal in length on dorsal and ventral surface. C2 widest claval segment.

*Mesosoma*: With imbricate sculpture, except lateral part of pronotum strigate, mesoscutum distinctly alveolate, upper mesepimeron alveolate, lower mesepimeron finely striate. Notauli distinct. Mesoscutellum shorter than mesoscutum, MSL/MSC about 0.7. Frenum distinct. Metanotum about one-third MSL, distinctly longer than frenum. Propodeum

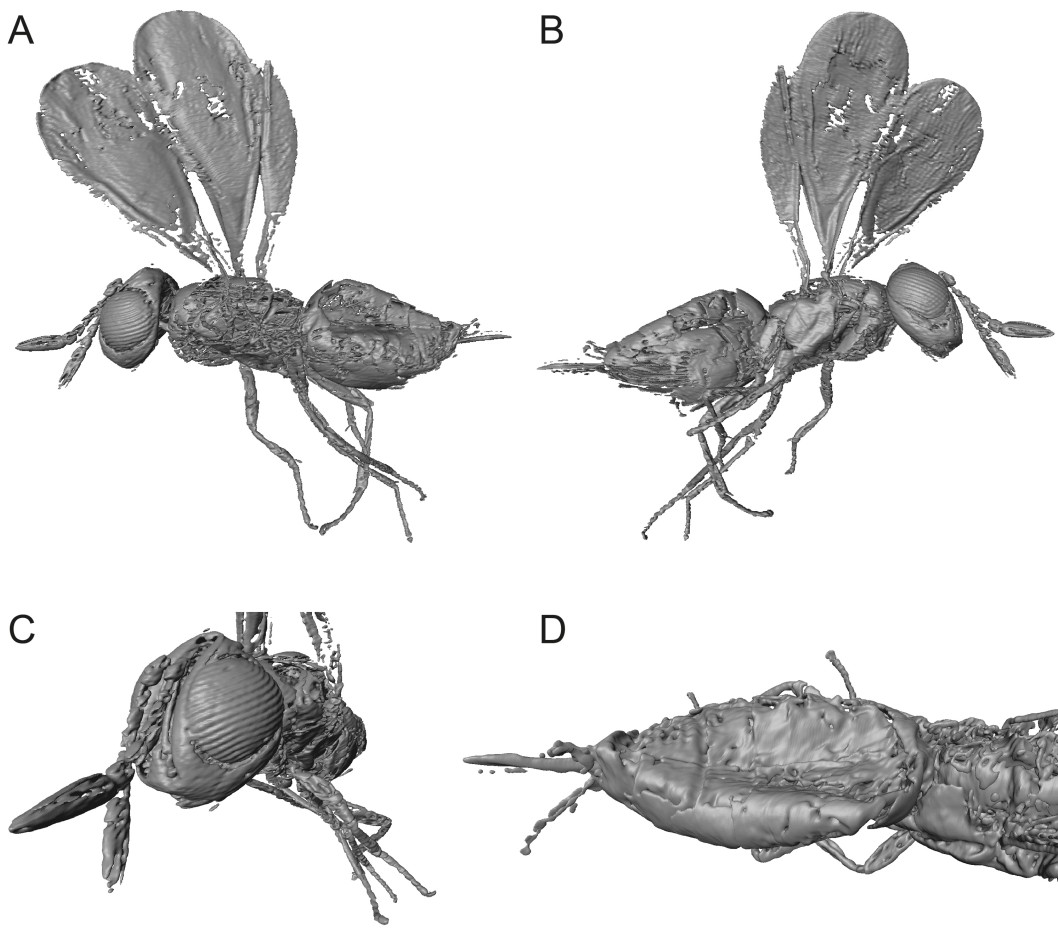

**Figure 3** Digital surface models of 3D reconstruction of *Spalangiopelta darlingi* sp. n., holotype, female. (A) Lateral habitus of the left side. (B) Lateral habitus of the right side. (C) Detail view of head and antennae. (D) Dorsal view of metasoma.

extremely short, only about one-third length of mesoscutellum. Prepectus broadly triangular, with imbricate sculpture.

Fore wing: broad, length 2.6 times maximum width, about 0.7 times body length. Hind margin rounded. Apical setae rather short and dense, length 0.6 times stigmal vein, less than one-sixth maximum width of fore wing. With 6 admarginal setae. Parastigma without hyaline break. Basal vein distinctly pigmented. Submarginal vein 1.3 times marginal vein. Marginal vein of uniform thickness, length subequal to postmarginal. Postmarginal vein about twice stigmal, PM/SV = 2.1. Stigmal vein slender and linear, making an angle of about 43 degrees with the postmarginal vein. Stigma significantly enlarged, uncus long with 3 or 4 sensilla.

Hind wing: narrow, length 6.4 times maximum width, with length of setae on apical and posterior margins 0.85 times width of hind wing.

*Metasoma*: Petiole transverse and rather inconspicuous. Gaster mostly shiny with fine reticulate sculpture on tergites, dark setae present laterally. T2 short, as long as T3-T4 combined. Laterotergite small, less than one-half height of gaster, with imbricate sculpture superficial. Ovipositor long, sheath extended distinctly beyond apex of metasoma, protruding from metasoma about 0.8 length of hind tibia. Length equal to *S. apotherisma* Darling & Hanson, 1986 (0.7) and the amber fossil *S. georgei*, the two known species with the longest ovipositor based on the ratio of the ovipositor sheaths' length and length of the hind tibia.

## Species description of *Spalangiopelta semialba* Moser, sp. n.

(Figs. 4–5)

LSID: urn:lsid:zoobank.org:act:10017E02-D18B-4731-94F6-3B5C2DBACE50

Holotype: ♀, Eocene Baltic amber, collection number ''SMNS BB-2852"

Type repository: SMNS.

The specimen is close to the surface of the amber piece. An air bubble on the left side of the specimen obscures the left mesopleuron, the propodeum and the anterior half of the metasoma. The preservation of the wings is rather poor with wing venation and pilosity hardly visible, only setae on margins are visible. Aside from the wasp, the amber piece contains three syninclusions, two mites (Acari) and one female Ceratopogonidae (Diptera) (Fig. 6).

Exact provenance unknown. Specimen is from the Krylov collection, purchased by the SMNS in 2008.

Distribution: Known only from the holotype.

Etymology: The specific name refers to the bicoloured head with the strikingly bright genal space.

Diagnosis: *Spalangiopelta semialba* differs from all fossil and extant *Spalangiopelta* species by the combination of a distinct malar sulcus, an inconspicuous petiole, and a wide fore wing (length about 3.2 times width). It is one of two known Eocene *Spalangiopelta* amber fossils from Baltic amber and it differs from the other fossil species in having a bright genal space, giving the head a bicoloured colouration. It is the smallest known species of *Spalangiopelta* with a body size of 0.55 mm.

Description: Female. Length, about 0.55 mm, slender in habitus. Head dark reddish-brown with faint iridescent reflections, genal space brighter, antennae light brown. Mesosoma concolorous dark reddish-brown with faint iridescent reflections, coxae and femora brown, tarsi lighter. Fore wing long, extended beyond apex of metasoma 0.8 times body length, hyaline. Metasoma concolourous dark reddish-brown.

*Head*: In frontal view rectangular, HW/HH: 0.5. With very fine, superficially imbricate sculpture, except glabrous vertex and ocular-ocellar areas. Line of black setae present along the parascrobal area and along the posterior vertex, lower face devoid of pubescence. Malar sulcus distinct and very long, over half eye height, malar sulcus 0.6 eye height. OOL about 0.5 POL.

Antenna: Radicle very short, about 0.15 scape length. Scape narrowly linear. F1short, length 0.65 width, F2-F3 slightly broader than long, F4-F5 subquadrate. Clava extremely elongate,

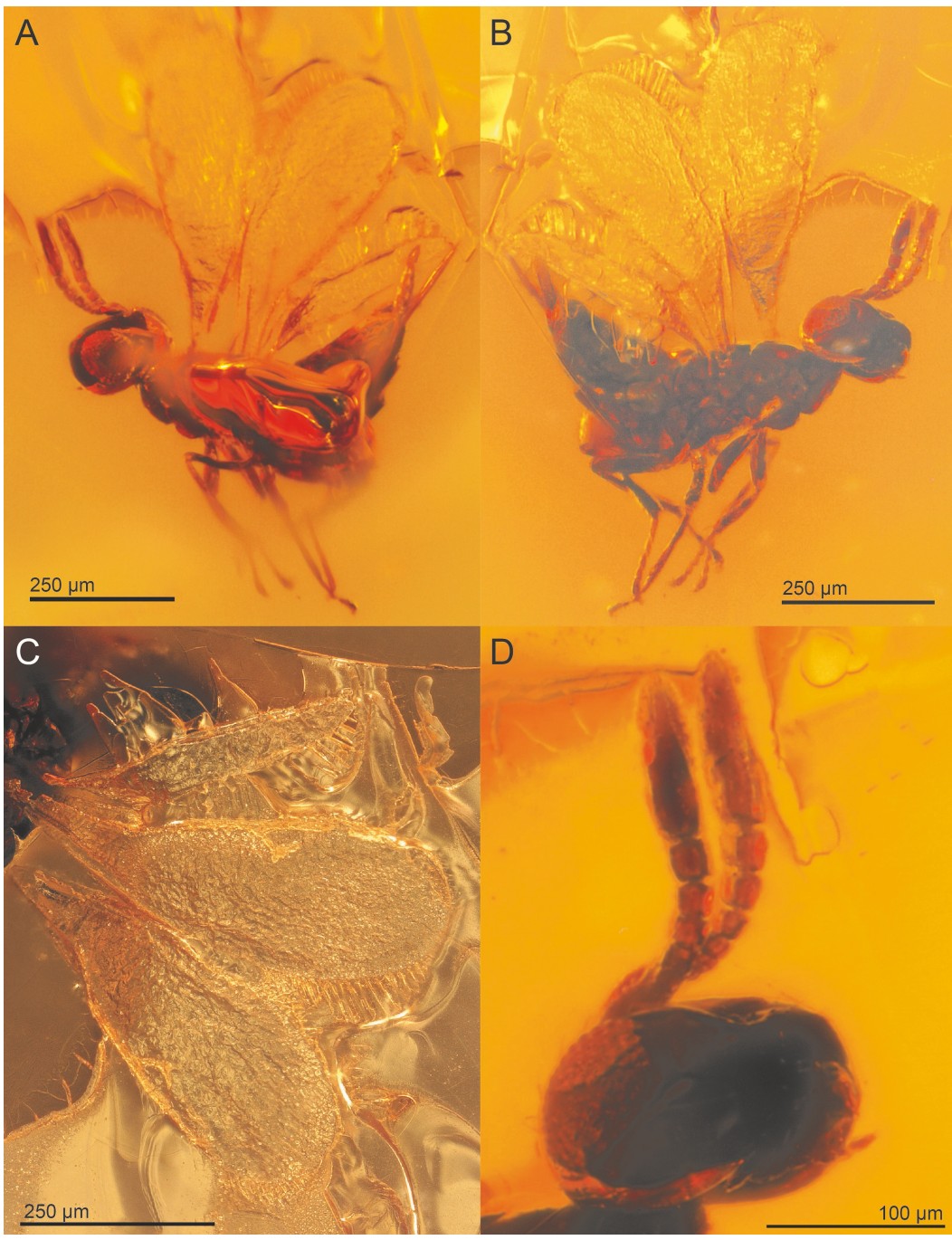

**Figure 4** *Spalangiopelta semialba* **sp. n., holotype, female.** (A) Lateral habitus of the left side. (B) Lateral habitus of the right side. (C) Detail image of wings. (D) Detail image of head with antennae.

slightly longer than F1-F5 combined, segments almost cylindrical, equal in length on dorsal and ventral surface. C1 widest claval segment, wider than F1-4, subequal in width to F5. *Mesosoma*: With imbricate sculpture. Notauli distinct. Mesoscutellum shorter than mesoscutum, MSL/MSC about 0.9. Frenum distinct (visible only in 3D reconstruction).

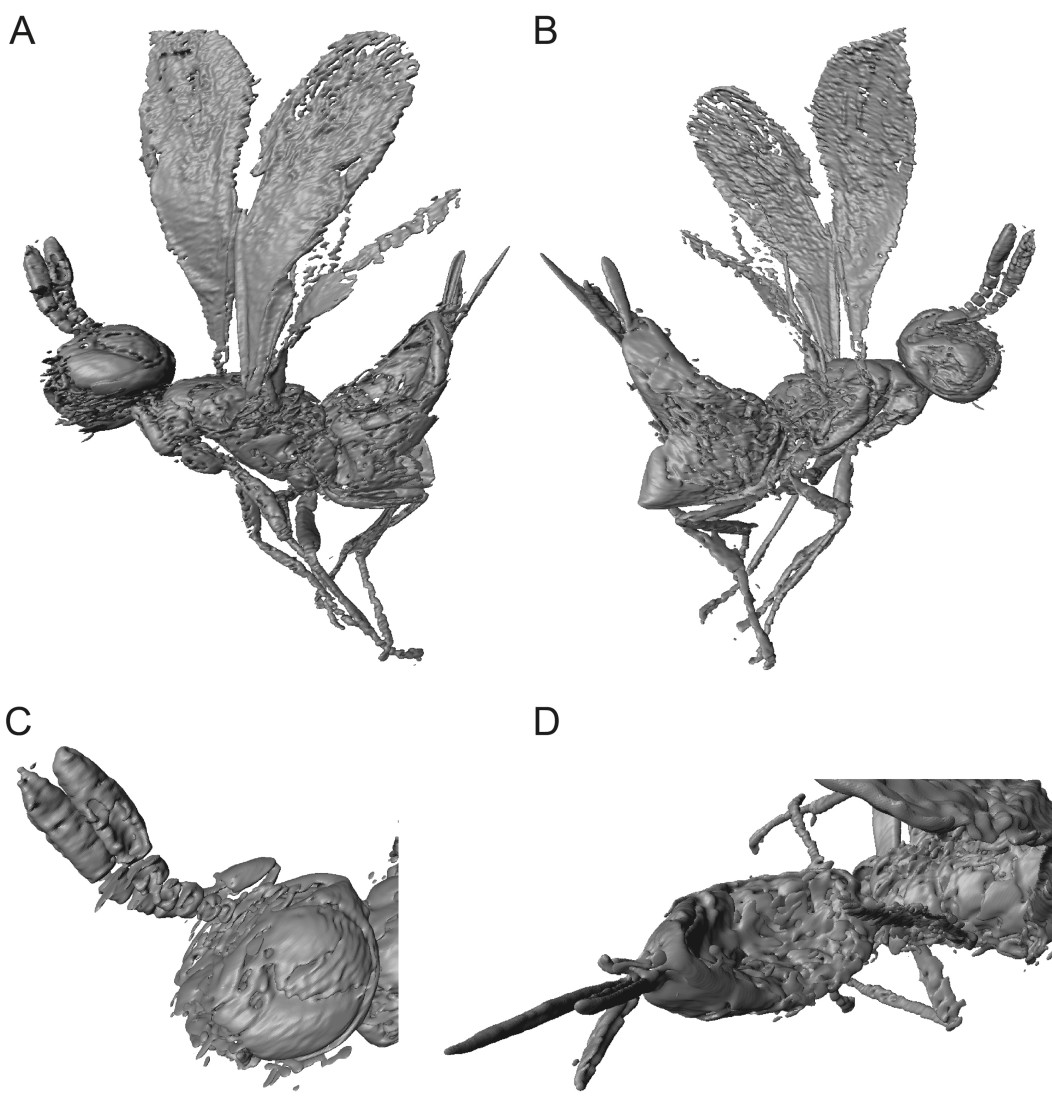

**Figure 5** Digital surface models of 3D reconstruction of *Spalangiopelta semialba* sp. n., holotype, female. (A) Lateral habitus of the right side. (B) Lateral habitus of the left side. (C) Detail view of head and antennae. (D) Dorsal view of posterior mesosoma and metasoma.

Metanotum very long, about 0.6 MSL, about one-half length of mesoscutellum along midline, longer than frenum. Propodeum short, only about one-half length of MSL. Prepectus broadly triangular, with imbricate structure.

Fore wing: broad, length about 2.9 times maximum width, of intermediate length, extended to apex of metasoma, about 0.8 body length. Hind margin only slightly expanded. Apical setae sparse, as long as stigmal vein. Admarginal setae or hyaline break are not visible due to the poor state of preservation. Submarginal vein 1.7 times length of marginal vein. Marginal vein of uniform thickness, length subequal to postmarginal. Postmarginal vein long, PM/SV = 3.3. Stigmal vein wide and straight, making an angle of about 55 degrees

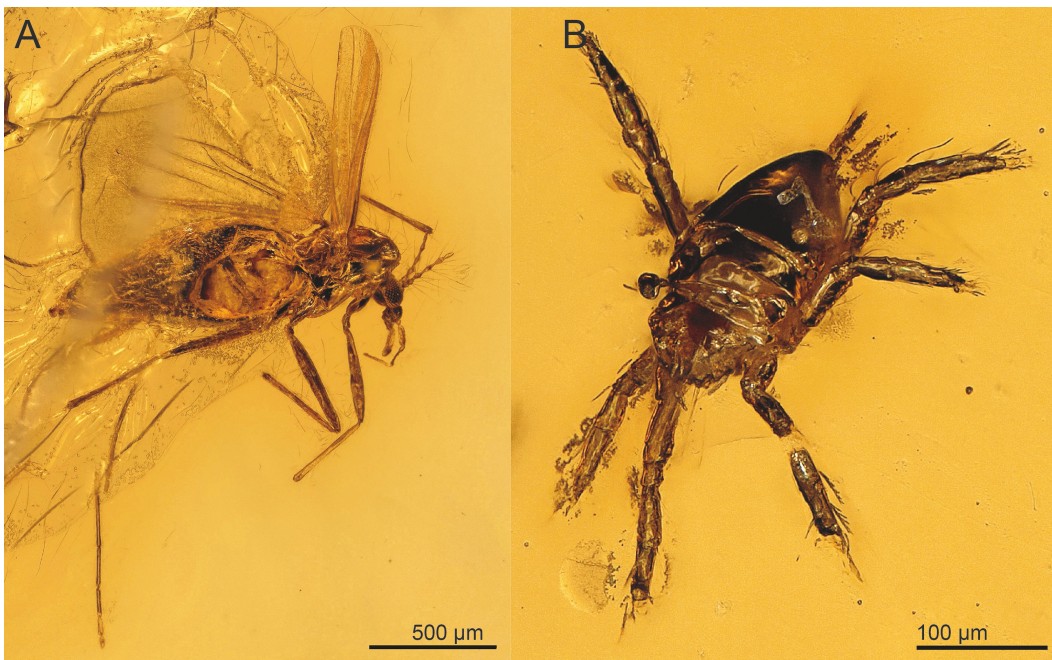

**Figure 6** **Syn-inclusions of *Spalangiopelta semialba* sp. n. in the amber piece "SMNS BB2852".** (A) Ceratopogonidae (Diptera). (B) Acari.

with the marginal and postmarginal vein. Stigma slightly enlarged, uncus long and broad, with 3 or 4 sensilla.

Hind wing: very narrow, length almost 9 times maximum width, with length of setae on apical and posterior margins subequal to width of hind wing.

*Metasoma*: Petiole inconspicuous. Gaster shiny with finely reticulate sculpture, laterally with scattered dark setae. T2 short, only as long as T3. Size and sculpture of laterotergite not clearly visible, thus not conclusively assessable. Ovipositor long, sheaths extended distinctly beyond apex of metasoma, protruded distance subequal to length of hind tibia.

## Phylogenetic analysis

In the unweighted phylogenetic analysis, both traditional search and implicit enumeration returned six trees that were equally parsimonious (length 58, consistency index 0.655; retention index 0.71, appendix C). In the strict consensus tree calculated from the equally parsimonious trees, the more distal species (i.e., *S. albigena*, *S. laevis*, *S. canadensis*, *S. alata* and *S. georgei*) are well-resolved, whereas the basal part of the tree is largely unresolved (Fig. 7). *Spalangiopelta dudichi* and *S. apotherisma* form a clade in the strict consensus tree. In all trees, *S. laevis* and *S. albigena* as well as *S. canadensis* and *S. alata* are returned as the two most distal sister clades. This cluster is well-supported as indicated by the high bootstrap values in the strict consensus tree. The Dominican amber fossil *S. georgei* is returned as sister group to this well-supported distal clade. However, this placement is not well supported and it would also contrasts with the cladogram in *Darling (1996)*, where *S. georgei* was returned as sister group to the most distal clade (*S. albigena* + *S. laevis*).

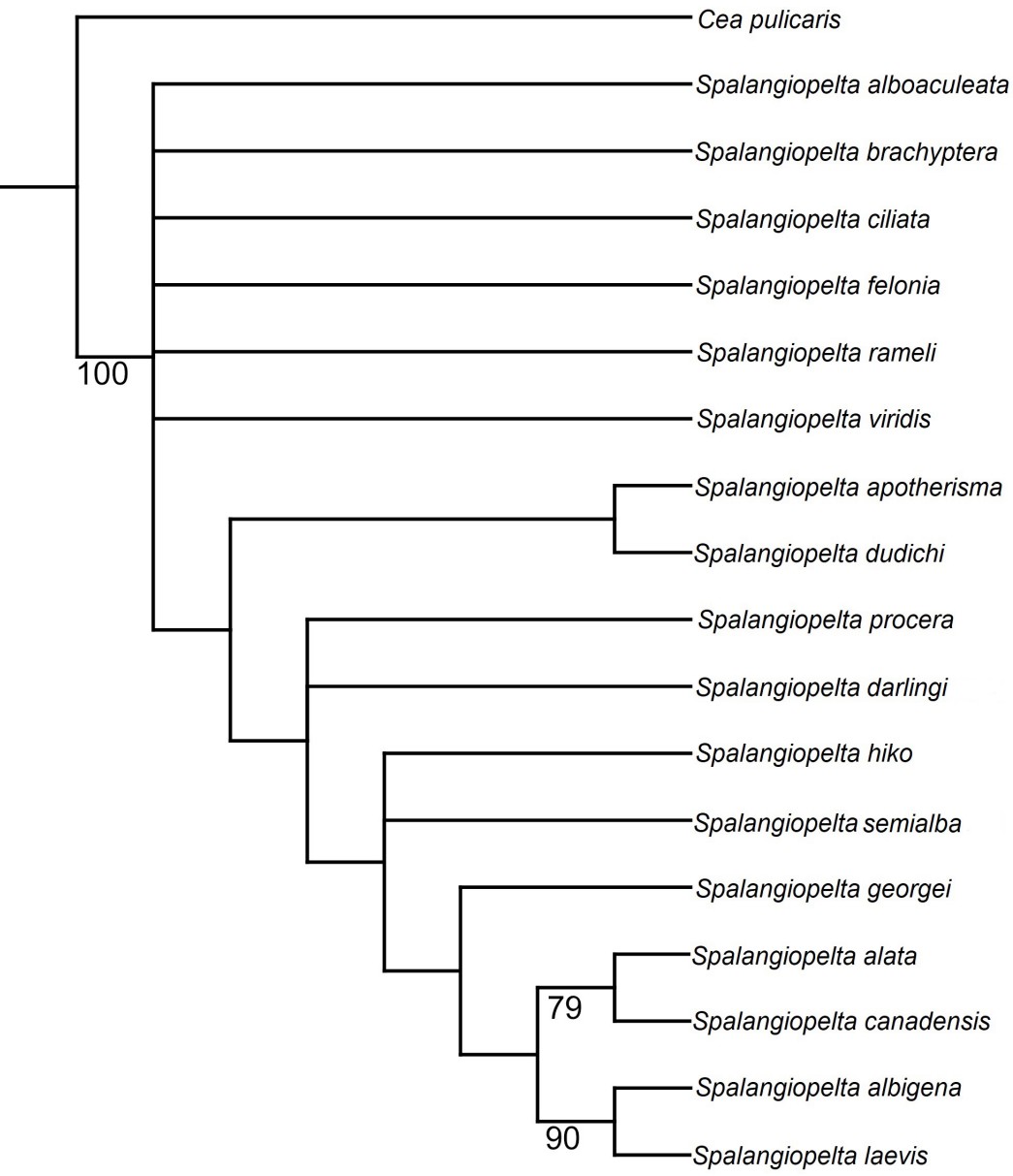

**Figure 7** **Equal weighting tree showing phylogenetic relationships between exant and fossil *Spalangiopelta* species based on female morphological characters.** Strict consensus calculated from 6 trees (length 58, consistency index 0.655; retention index 0.71, 18 taxa, 22 characters). Only bootstrap values over 70 are shown. Bootstrap values represent absolute frequencies based on 1,000 replicates.

In the strict consensus tree, the Baltic amber fossil *S. darlingi* forms a polytomy with *S. procera*. Its exact placement is unresolved in that it is either returned as sister group to the *S. ducichi* + *S. apotherima* clade or as sister taxon to *S. hiko*. *Spalangiopelta semialba*, the other Baltic amber fossil, forms a polytomy with *S. hiko*.

Phylogenetic analyses using implied weighting with concavity constant values lower than $K = 4$ return multiple trees of length 59 or 60, whereas concavity constant values

higher than $K = 4$ produced three trees of length 58. Therefore, the preferred tree topology presented here is the result of implied weighting of characters with a concavity constant of $K = 4$ (Fig. 8). Tree topology largely corresponds to the strict consenus (Fig. 7) but shows a better resolution along the base. *Spalangiopelta ciliata* is retrieved as sister group of the remaining *Spalangiopelta* species.

The implied weighting tree retained in this analysis largely corresponds to that of *Darling (1996)*. However, there are three major differences in topology: (1) In the preferred tree topology of this study, *S. ciliata* constitutes the most basal species of *Spalangiopelta*. (2) *S. felonia* and *S. brachyptera* no longer cluster with *S. alboaculeata* as their sister group. Instead, they represent separate lineages. (3) The most distal part of the cladogram is represented by two synapomorphic clades, (*S. albigena* + *S. laevis*) and (*S. alata* + *S. canadensis*) with *S. georgei* as sister group to this most distal clade (Fig. 8).

The two Baltic amber fossils are closely related to each other and to the Western European *S. procera* as well as *S. hiko*, which has been found only in Japan (Fig. 8). The two new amber fossil species have little influence on the phylogeny. With *S. darlingi* and *S. semialba* removed from the phylogenetic analysis, topology changes only slightly. *S. georgei* becomes the sister group of the most distal clade of the two Caribbean species *S. laevis* + *S. albigena*, and *S. apotherisma* no longer forms a monophyletic clade with *S. dudichi*.

## DISCUSSION

### Character evolution and functional morphology of the mesopleuron in *Spalangiopelta*

The distinct configuration of the hind margin of the mesopleuron, which is distinctly raised and partly covering the metapleuron, separates *Spalangiopelta* from the other genera in the subfamily Ceinae. This feature was hypothesized to facilitate flexible movement of the sclerites, thereby giving the mesosoma a more arched profile (*Darling, 1991*). This character is present in both fossils, although it is more pronounced in *S. darlingi*. However, this character is not unique to the genus *Spalangiopelta*. In a comprehensive phylogenetic analysis of Chalcidoidea, this character was found in a range of taxa (character 111 "Mesepimeron relative to metapleural/propodeal complex" in *Heraty et al., 2013*). It was found in multiple species of Pteromalidae, Eulophidae, Eurytomidae and Tetracampidae and occasionally in Eriaporidae, Leucospidae and Perilampidae. Within the Pteromalidae examined by *Heraty et al. (2013)*, this character is found in the subfamilies Cleonyminae, Leptofoeninae, Diparinae and Spalangiinae. Although these groups are morphologically distinct, a connection to forest habitats is common to many species within these subfamilies: The development of the majority of Cleonyminae and Leptofoeninae takes place inside wood (*Vilhelmsen & Turrisi, 2011*). Diparinae are associated with leaf litter on the forest floor (*Desjardins, 2007*). Some Spalangiinae specimens were found in forest habitats (*Gibson, 2009*). Therefore, it seems reasonable to acknowledge the hypothesis that *Spalangiopelta* females are associated with leaf litter habitats (*Darling, 1991*). In addition, the wing reduction in females of *S. brachyptera* and the continuous variation in wing length reported in *S. felonia*, which have been found in large numbers in Berlese funnel extractions

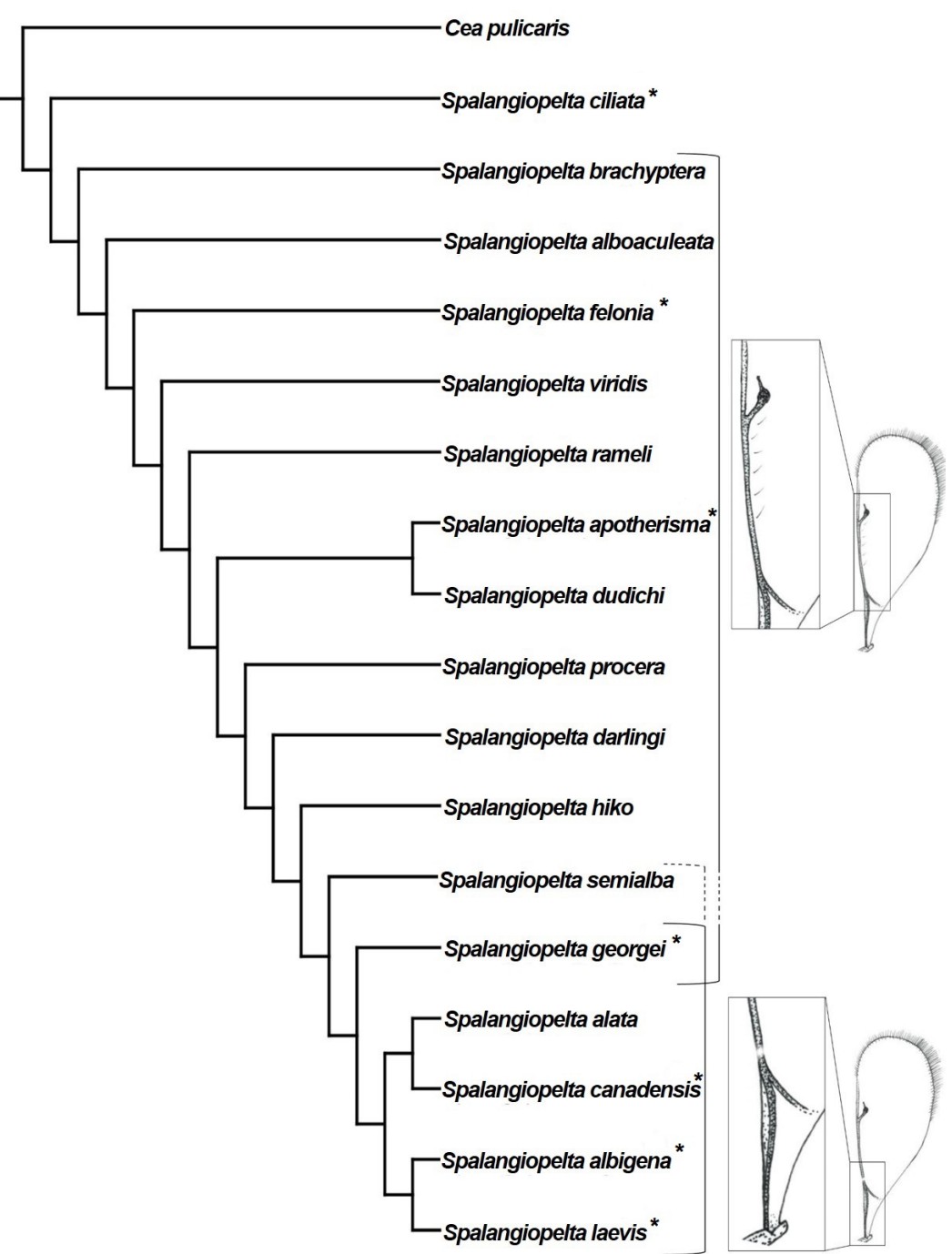

**Figure 8   Implied weighting tree ($k = 4$) showing phylogenetic relationships between exant and fossil** *Spalangiopelta* **species based on female morphological characters.** Tree length 58, 18 taxa, 22 characters. Resolution of the tree is based on implied weighting. Asterisks (*) annotate New World species. Insets map the distribution of admarginal setae (above) and hyaline break (below).

of the needle mat in a Douglas fir forest (*Pseudotsuga menziesii* (Mirb.) Franco (Pinaceae)), place these *Spalangiopelta* species into spatially restricted forest floor habitats (*Darling & Hanson, 1986*; *Masi, 1922*). This reasoning is in line with the general rarity of *Spalangiopelta* specimens in collections. The presence of the raised mesopleuron in the newly described fossil species, as well as their overall morphological similarity to extant species, suggest that these species might have lived in similar forest habitats 35 to 43 million years ago. Further, a wooded habitat would increase the chance for a specimen to turn into an amber fossil.

Within Pteromalidae, *Macroglenes gramineus* (Haliday 1833), *Spalangia nigroaenea* Curtis 1839 and *Peckianus* sp. Bouček 1975 also share the characteristic configuration of the mesopleuron that partly overlaps the metapleuron. Except for the Australian *Collessina pachyneura* Bouček 1975 and the monotypic genus *Peckianus*, which is known only from Canada and Brazil, all other species listed above are associated with various Dipteran families as hosts (*Noyes, 2019* and references therein).

## Character evolution and functional morphology of admarginal setae and the hyaline break in *Spalangiopelta*

The most striking character that unites the distal species in this cladogram is the presence of an unpigmented spot in the parastigma of the fore wing termed "hyaline break", (character 1, Fig. 8). With the exception of *S. alata*, all distal species in the cladogram that possess a hyaline break in the fore wing are New World species. The presence or absence of a hyaline break and admarginal setae (character 7) within the genus *Spalangiopelta* has been a subject of discussion in other studies (*Darling, 1991*; *Darling, 1996*). Every macropterous species of *Spalangiopelta* has either a hyaline break or admarginal setae close to the marginal vein on the underside of the fore wing. The only exceptions are *S. ciliata*, which has neither, and the Dominican amber fossil *S. georgei*, which has admarginal setae and a hyaline break (*Darling, 1996*). *Darling (1991)* considers the absence of either character in *S. ciliata* the result of the regular above-ground activity of the species, which would explain why *S. ciliata* is found in Malaise traps regularly. Of the two fossils described herein, *S. darlingi* has six admarginal setae whereas in *S. semialba*, the state of preservation prevents the reconstruction of this character with certainty. From the pattern in the phylogeny it cannot be inferred whether *S. semialba* had admarginal setae, a hyaline break or both (Fig. 8). If *S. semialba* had admarginal setae, this would imply that the distal clade ((*S. alata* + *S. canadensis*) + (*S. albigena* + *S. laevis*)) share a common ancestor that lost admarginal setae and developed a hyaline break. This process might have occurred roughly 15 to 20 million years ago considering the age of the Dominican amber fossil *S. georgei*, which might represent a transitional stage (*Darling, 1996*). If like in *S. georgei* admarginal setae and a hyaline break were present in the Baltic amber fossil *S. semialba*, the process of losing admarginal setae and developing a hyaline break could be dated back much further.

Both wing characters have been suggested to help *Spalangiopelta* keep the wings folded back to facilitate movement in restricted habitats such as leaf litter (*Darling, 1991*). In a study that placed Ceinae as sister group of Diparinae, *Desjardins (2007)* was able to observe "many diparine specimens […] in which the setae appear to hold the hind wing in place while they are folded against the body", thereby potentially verifying the hypothesis

proposed several years earlier. The author acknowledges that the presence of admarginal setae in some species of Ceinae as well as Diparinae does not necessarily denote relatedness but could indicate homoplasy based on the similarity of the habitats of the hosts. The absence of admarginal setae in *S. ciliata*, which was resolved as sister group to all other *Spalangiopelta* species in the phylogenetic analysis, seems to confirm this view.

Based on the hypothesis that the hyaline break serves an identical purpose as the admarginal setae, *Darling (1991)* considered these wing characters mutually exclusive. However, there are several species of pteromalid wasps that share both wing characters, including *C. pachyneura*, *M. gramineus*, *S. nigroaenea*, *Pteromalus albipennis* Walker 1835 and an unidentified species of the genus *Peckianus* (characters 147 and 157 of *Heraty et al., 2013*). As both wing characters can be found not only in females but also in males of *Spalangiopelta*, it appears that neither character is associated directly with host location. Admarginal setae and a hyaline break do occur mutually in at least five species of Pteromalidae. The results of the phylogenetic analysis presented here provide further evidence for the mutual presence of admarginal setae and a hyaline break in *S. georgei* and potentially *S. semialba*. They contradict the postulation by *Darling (1991)* that the mutual exclusivity is the result of identical functions of both characters. As *Darling (1991)* stated, the hyaline break is "an example of a fenestra or bulla, a weakened area of a vein that usually marks an area where a fold or flexion line crosses the vein". Fenestrae occur where a flexion line crosses a wing vein and are associated with complex changes of wing shape in flight (*Danforth & Michener, 1988*; *Wootton, 1979*). Therefore, we suggest that the hyaline break found in the distal clade of *Spalangiopelta* presented here might be an adaptation to flying rather than to moving in confined habitats.

## CONCLUSIONS

We add two new species to the fossil record of Chalcidoidea (Hymenoptera) from Baltic amber and establish a minimum age of the genus *Spalangiopelta* of 35 to 43 million years (upper Eocene). Further, we present a cladogram within which we place the fossils based on the cladistic analysis of 22 morphological characters. The new identification key to the genus includes all extant and fossil females of *Spalangiopelta*.

The description is complemented by 3D models reconstructed from μCT scans. The models allow for a pivotable view and thus eliminate physical barriers such as refraction or reflections that would otherwise obscure the view on specific parts of the inclusion depending on the cut of the amber piece. Inclusions such as air bubbles or plant particles that conceal potentially important structures of the specimen can also be removed digitally. This method has proved an invaluable addition to classic techniques in the description of amber fossils (*Dierick et al., 2007*; *Faulwetter et al., 2013*; *van de Kamp et al., 2018*).

Coding additional morphological characters and including genetic data will further improve the resolution of the phylogenetic analysis given here. Additional characters of males could elucidate aspects of the phylogenetic placement that have yet remained unclear such as the size of the metascutellum or the shape of the fore wing. To obtain these characters, molecular data could be utilized to decisively match females with males

of *Spalangiopelta*. Increasing the number of *Spalangiopelta* specimens in collections could prove productive to assess the range of absolute measurements in certain characters within one species as well as to classify the variation in characters such as wing length in *S. felonia*. This increase could be achieved by targeted biodiversity assessments in promising habitats such as forest floors. In order to validate any hypotheses regarding the functional morphology of prominent structures such as the hyaline break, the admarginal setae and the raised mesopleuron, the observation of live specimens could yield presently unknown insights into functional morphology, behaviour and host range of the genus *Spalangiopelta*.

**Abbreviations**

| | |
|---|---|
| **A1-A3** | anelli 1-3 |
| **C1-C3** | claval segment 1-3 |
| **F1-F5** | funicular segment 1-5 |
| **HH** | head height |
| **HW** | Head width |
| **LOL** | lateral ocellar line |
| **MSC** | mesoscutum |
| **MSL** | mesoscutellum |
| **OOL** | ocular-ocellar line |
| **POL** | posterior ocellar line |
| **PM** | postmarginal vein |
| **SV** | stigmal vein |
| **T1-T7** | metasomal tergites 1-7 |
| **SMNS** | Staatliches Museum für Naturkunde Stuttgart/ State Museum of Natural History Stuttgart |

# ACKNOWLEDGEMENTS

We thank Tanja Schweizer (SMNS) for technical assistance in the preparation of the amber pieces. We would also like to show our gratitude to Michael Haas and Milan Pallmann (both SMNS) for support in operating imaging hardware and software. We thank Marcus Zuber and Tomás Faragó (both KIT) for their assistance during the tomographic experiment. We acknowledge the KIT light source for provision of instruments at their beamlines and we would like to thank the Institute for Beam Physics and Technology (IBPT) for the operation of the storage ring, the Karlsruhe Research Accelerator (KARA). We thank André Nel, Mircea-Dan Mitroiu and one anonymous reviewer for their time and effort in helping to improve an earlier version of this manuscript.

## Funding

Funding for a research stay of RMB at SMNS was provided through a National Science Foundation grant (NSF DEB-1555808). The funders had no role in study design, data collection and analysis, decision to publish, or preparation of the manuscript.

## Grant Disclosures

The following grant information was disclosed by the authors:
National Science Foundation: NSF DEB-1555808.

## Competing Interests

The authors declare there are no competing interests.

## Author Contributions

- Marina Moser performed the experiments, analyzed the data, prepared figures and/or tables, authored or reviewed drafts of the paper, and approved the final draft.
- Roger A. Burks and Lars Krogmann conceived and designed the experiments, analyzed the data, authored or reviewed drafts of the paper, and approved the final draft.
- Jonah M. Ulmer and John M. Heraty analyzed the data, authored or reviewed drafts of the paper, and approved the final draft.
- Thomas van de Kamp performed the experiments, analyzed the data, authored or reviewed drafts of the paper, and approved the final draft.

## Data Availability

MicroCT Data are available at Morphosource: https://www.morphosource.org/projects/0000C1156.

- *Spalangiopelta darlingi* surface model; Media 000165854; ark:/87602/m4/M165854; MorphoSource DOI 10.17602/M2/M165854

- *Spalangiopelta darlingi* CT ImageSeries; Media 000165855; ark:/87602/m4/M165855; MorphoSource DOI 10.17602/M2/M165855

- *Spalangiopelta semialba* surface model without air bubble; Media 000165856; ark:/87602/m4/M16585; MorphoSource DOI 10.17602/M2/M165856

- *Spalangiopelta semialba* surface model with air bubble; Media 000165858; ark:/87602/m4/M165858; MorphoSource DOI 10.17602/M2/M165858

- *Spalangiopelta semialba* CT ImageSeries; Media 000165857; ark:/87602/m4/M165857; MorphoSource DOI 10.17602/M2/M165857

The holotype of *Spalangiopelta darlingi* is deposited at SMNS (Accession number: SMNS BB-2851). The holotype of *Spalangiopelta semialba* is deposited at SMNS (Accession number: SMNS BB-2852).

## New Species Registration

The following information was supplied regarding the registration of a newly described species:

Publication LSID:
urn:lsid:zoobank.org:pub:F3A4D890-3480-41D2-B5A4-7C18E047C920
*Spalangiopelta darlingi* sp. n. LSID:
urn:lsid:zoobank.org:act:91C3491D-167E-438E-A336-36534722525D
*Spalangiopelta semialba* sp. n. LSID:
urn:lsid:zoobank.org:act:10017E02-D18B-4731-94F6-3B5C2DBACE50.

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
