# Peer review of "Taxonomic description and phylogenetic placement of two new species of Spalangiopelta (Hymenoptera: Pteromalidae: Ceinae) from Eocene Baltic amber"

_PeerJ, doi:10.7717/peerj.10939_

## Round 0.1 · original submission · Minor Revisions

In light of three independent reviews, that report a good manuscript and very careful research was done, I believe this manuscript deserves to be published after a minor review. Please consider the suggestions made by all reviewers, considering their suggested improvements and resubmit the manuscript by January 30th 2021.

·

Basic reporting

all ok !!

I do see clearly the impact of the ct scans, they are not very precise, can you explain

Experimental design

all ok

Validity of the findings

all ok, nice finding for a rather poorly known group

Additional comments

whereas the other six species are Nearctic or Neotropic species
remove last 'species'

Character 1: Hyaline break in parastigma. 0: absent, 1: present. The third character
143 state (“partial”) in Darling (1996) was removed
please, precise to which remaining state absent vs present they are transferred, and why ?

What mean Presence of short-winged females.
short compared to what, can you precise ? please

Character 17: Shape of forewing. 0: narrow, length greater than 3.5 times width, 1:
199 wide, length less than 3.5 times width
what about if it very close to 3.5 ?

Fore wing with transparent region (“hyaline break”) in parastigma ......................3
271 — Fore wing with large admarginal setae below marginal vein..................................4
the two states are not opposed ?

the diptera in syninclusion is a female Ceratopogonidae

Reviewer 2 ·

Basic reporting

Clear and unambiguous, professional English used throughout - for the most part. Problem areas highlight in the pdf returned with this review. The literature citations are complete as far as I can tell.

Experimental design

The 3D models constructed based on the CT data add nothing to the paper. They do not clarify any aspects of the morphology as presented and if anything detract from the other figures in the manuscript. The habitus drawing (Fig. 1) is minimally useful but should show the admarginal setae present in Fig. 2.

Validity of the findings

I am not convinced that two species of Spalangiopleta are represented in this study. S. darlingi is well supported by Figure 2 but the status of the specimen described as S. semialba is less clear -- the white genae are certainly not prominent in Figure 4. And the distinctive mesopleuron of the genus is not apparent in Figure 4. The Diagnosis of S. semialba suggest to me the possibility that this specimen is not referable to Spalangiopelta.

Additional comments

See annotated pdf for suggestions on wording and other comments.

Figure 1. The habitus drawing is minimally useful but should show the admarginal setae present in Fig. 2.
Figure 6. This should be deleted. The "syninclusions" add nothing to the paper.
Caption, figure 8. Add (above) and (below) to the inset text.
Table head. What are Amira measurements. Explain here or in the text.

Annotated reviews are not available for download in order to protect the identity of reviewers who chose to remain anonymous.

·

Basic reporting

The manuscript is an important addition to the knowledge of a rather poorly studied group of chalcid wasps. It adds two new fossil species, keys all known species of Spalangiopelta, and provides a cladistic analysis of the genus. The most important area where the authors should pay attention is the key to species, where in several couplets the two parts do not match entirely, i.e. some characters are mentioned in the first part of the couplet, but are not repeated in the second part, or vice versa. The authors should also clarify the use of the terms 'absent' and 'indistinct' as different, which can be confusing. Regarding the phylogenetic analysis, although it is not very informative regarding the basal part of the tree, it provides some interesting correlations between some morphological characters.

Experimental design

No comment

Validity of the findings

The two new species are well described, allowing comparisons with previously described species. They are well illustrated and benefit from the cutting edge technology of micro CT.

Additional comments

Please see and address all suggestions and comments in the manuscript.

---

## Round 0.2 · accepted · Accept

Dear Dr. Krogmann,

After the reviewer who requested major reviews to the previous version of your manuscript indicated that all suggested changes were made in this version of your manuscript, I am pleased to inform you that your manuscript is now accepted for publication in PeerJ. Congratulations!

·

Basic reporting

All indicated changes/comments have been addressed.

Experimental design

All indicated changes/comments have been addressed.

Validity of the findings

All indicated changes/comments have been addressed.

Additional comments

All indicated changes/comments have been addressed, thank you.